# *SeLoRA*: Self-Expanding Low-Rank Adaptation of Latent Diffusion Model for Medical Image Synthesis

## Abstract

The persistent challenge of medical image synthesis posed by the scarcity of annotated data and the need to synthesize "missing modalities" for multi-modal analysis, underscored the imperative development of effective synthesis methods. Recently, the combination of Low-Rank Adaptation (*LoRA*) with latent diffusion models (LDMs) has emerged as a viable approach for efficiently adapting pre-trained large language models, in the medical field. However, the direct application of *LoRA* assumes uniform ranking across all linear layers, overlooking the significance of different weight matrices, and leading to sub-optimal outcomes. Prior works on *LoRA* prioritize the reduction of trainable parameters, and there exists an opportunity to further tailor this adaptation process to the intricate demands of medical image synthesis. In response, we present *SeLoRA*, a Self-Expanding Low-Rank Adaptation module, that dynamically expands its ranking across layers during training, strategically placing additional ranks on crucial layers, to allow the model to elevate synthesis quality where it matters most. Our analysis shows that *SeLoRA* strikes the best balance between synthesis quality and training efficiency. The proposed method not only enables LDMs to fine-tune on medical data efficiently but also empowers the model to achieve improved image quality with minimal ranking. The code of our *SeLoRA* method is publicly available at this link.

## 1 Introduction

Foundation models Tu et al. (2024); Huang et al. (2023); Zhou et al. (2023) are increasingly gaining traction in medical imaging, offering a new paradigm for data processing and analysis. While most foundational models are trained with large natural-image datasets Tu et al. (2024), such as *ImageNet*, the shortage of medical images is increasingly problematic (Deng et al., 2009). Medical image synthesis presents a valid approach to address this issue by generating synthetic images to expand and enhance scarce image datasets. Challenges in medical image synthesis have been widely explored, leading to various proposed methods. These models are typically trained from scratch for a single modality like brain MRI Dalmaz et al. (2022), lung CTsMendes et al. (2023), cataract surgery samples Frisch et al. (2023), and others. However, training models from scratch requires long training times and can lead to performance limitations due to dataset size.

Recognizing the success of pre-trained foundational models on natural images, recent works have shifted towards adapting these models, such as stable diffusion Rombach et al. (2022), to enable more efficient training for medical image synthesis. For example, leveraging text-based radiology reports as a condition to incorporate detailed medical information, and fine-tuning latent diffusion models has achieved significant performance gains (Chambon et al., 2022a;b). Further exploration involves the incorporation of parameter-efficient fine-tuning (PEFT) methods, which not only makes fine-tuning more efficient but also demonstrates superior performance compared to full fine-tuning (Dutt et al., 2023). In this context, we place a particular emphasis on applying the Low Rank Adaptation (*LoRA*) Hu et al. (2022) method, a type of PEFT method, for fine-tuning stable diffusion in medical image synthesis.

Figure 1: Training illustration of a single *SeLoRA*. *SeLoRA* behaves similarly to a basic *LoRA* during training. However, it is tested for the expanded rank every $t$ step and is progressively expanded if the FI-Ratio exceeds the desired threshold.

Originally designed for adapting large language models, *LoRA* hypothesizes that the weight matrix updated during fine-tuning exhibits a low 'intrinsic rank'. Therefore, it proposed to use the multiplication of two trainable low-rank decomposition matrices to mimic the weight update for a specific task, expressed as follows:

$$W = W_0 + AB, \tag{1}$$

The weight matrix, $W^{d_{in} \times d_{out}}$, is updated by adding the product of trainable low-rank decomposition matrices, $A \in \mathbb{R}^{d_{in} \times r}$ and $B \in \mathbb{R}^{r \times d_{out}}$ to the frozen original weight matrix $W_0^{d_{in} \times d_{out}}$. The rank, $r$, is deliberately chosen to be significantly smaller than the dimension of the original weight. Consequently, the trainable parameters under this configuration constitute a fraction of $W_0$'s parameter count, achieving parameter-efficient fine-tuning.

The design of *LoRA* introduced the challenge of selecting optimal rank. Small ranks yield suboptimal performance, while large ranks escalate parameter count, and searching for the optimal rank for each individual *LoRA* on different layers is computationally expensive. In most large language models, where *LoRA* is primarily applied, the Transformer-like architecture typically have similarsized weights across layers. Therefore, *LoRA* proceeds with having uniform rank across all layers, assuming that the updated weights also have similar ranks throughout. This simplifies the challenge of selecting optimal by reducing it from searching an optimal rank for each layer to selecting a single rank for the entire model. However, given our context of applying *LoRA* to Stable Diffusion, the bottleneck structure of the Denoising U-Net within Stable Diffusion introduces a diverse range of weight matrix shapes. Using a uniform rank becomes problematic, as the inherent rank of these weight matrices can vary significantly - larger matrices may require higher ranks, while smaller ones may need lower ranks. Consequently, selecting a uniform rank for *LoRA* that is applied on LDMs may lead to suboptimal results, ultimately compromising the quality of synthesized images.

In addressing the challenge of rank selection and aiming to achieve superior synthesized image quality with minimal trainable parameters, we draw inspiration from the concept of self-expanding neural network Mitchell et al. (2024). Departing from approaches of setting a predefined uniform rank, our approach advocates for dynamically expanding the rank of *LoRA* to better align with the unique needs of each layer. As a result, in our work, we present Self-Expanding Low-Rank Adaptation (SeLoRA), akin to *LoRA*'s structure but distinguished by the dynamic growth of ranks guided by Fisher information during training. This enables *SeLoRA* to flexibly adapt to the inherent characteristics of each layer, guaranteeing enhanced medical image synthesis quality while minimizing challenges related to rank adjustments.

## 2 RELATED WORK

Low-rank adaptation has gained significant attention due to its memory-efficient nature, enabling a broader community to fine-tune increasingly large models. Building upon *LoRA*, several research directions have emerged, primarily focusing on reducing the number of trainable parameters through techniques such as quantization, random weights, various product operations, and adaptive rank selection. These methods are relevant to our work, as we aim not only to enable adaptive rank in *LoRA* but also to minimize the overall parameter count.

*QLoRA*, for example, reduces the memory of *LoRA* weights by quantizing them to 4 bits using two methods: 4-bit NormalFloat quantization and Double Quantization. This results in a fourfold reduction in memory usage for *LoRA* weights (Dettmers et al., 2023). *VeRA* replaced the trainable low rank matrices with an frozen random matrices, and uses two trainable scaling vectors between and after these frozen random matrices to adapt to the fine-tuning tasks, further reducing the parameter count (Kopiczko et al., 2024). *Tied LoRA* builds on *VeRA* by exploring various combinations of shared and frozen weights for both scaling vectors and low-rank matrices (Renduchintala et al., 2024). Other approaches replace the original low-rank matrix multiplication with alternative operations to reduce the parameter count even further. *KronA*, for instance, uses the Kronecker product to substitute the standard matrix multiplication in *LoRA* (Edalati et al., 2022), while *LyCORIS* incorporates both the Kronecker and Hadamard products to create a its variant of *LoRA* (Yeh et al., 2024). All of these methods aim to reduce *LoRA*'s memory requirements through different techniques. Its important to note that these approaches are complementary to our proposed methods, suggesting greater efficiency when our approaches combined with these existing techniques.

**Adaptive Rank Selection**    A particularly relevant line of research to our method is adaptive rank selection, which aims to dynamically select the optimal rank for *LoRA* during training. This is especially pertinent as it addresses the critical issue of rank selection problem. *Noah* directly injects *LoRA*, *adapter*, and *bitfit* into an attention layer, using evolutionary search to identify the optimal rank and configuration (Zhang et al., 2022; Zaken et al., 2022; Houlsby et al., 2019). Similarly, *Generalized LoRA* introduces a super-net that can mirror the structure of *LoRA*, scalar weights, or bias weights, forming a generalized version of *LoRA*. It also pursues adaptive rank selection through through evolutionary search during training (Chavan et al., 2023). While these approaches can effectively find an optimal rank through evolutionary search, they come with trade-offs. The repeated search process can significantly extend training time, and in some cases, the search explores high-rank matrices, which inherently have a large number of parameters. This increased parameter count can lead to higher memory usage, counteracting *LoRA*'s original goal of memory efficiency.

**Rank Pruning**    Rank pruning first sets an upper bound to the rank of *LoRA*, and gradually prunes the rank of each *LoRA* adapter during training to reduce the parameter count and search for the optimal rank. *DyLoRA* uses a nested dropout loss to sort and train the rows of the low rank matrices, placing higher-information rows at lower row indices. After training, the low-rank matrices can be pruned to a user-defined rank without losing significant information, since the most informative rows at the lower row indices are preserved (Valipour et al., 2022). *AdaLoRA* assigns an importance score for each row of the weight matrix, evaluated based on the magnitude of singular values, thereby allows pruning of less important ranks during training (Zhang et al., 2023). *SoRA* introduces a specially designed gating function that zeros out redundant ranks during training, effectively reducing the rank and improving efficiency.(Ding et al., 2023). While these pruning-based methods effectively achieve adaptive rank across layers, the fundamental challenge of rank selection remains only partially addressed. By introducing a fixed upper bound as the initial rank for the trainable low-rank matrices, these methods may limit the exploration of more optimal configurations.

To address the limitations of fixed upper bounds while selecting optimal rank across layers, we chose to explore the possibility of rank expansion during training. A parallel work, *ALoRA*, also introduces rank expansion but with different rules, expanding rank only when pruning is not triggered Liu et al. (2024). Our approach, however, places greater emphasis on how to increase the rank strategically. By using Fisher information as a guideline, we aim to expand the rank in each layer more effectively, ensuring that the rank growth aligns with the specific needs of the model.

## 3 SELORA

In addressing the challenge of rank selection and maintaining superior performance with minimum rank, we propose *Self-Expanding Low-Rank Adaptation* (*SeLoRA*). The general idea of *SeLoRA* is to initialize the trainable low-rank decomposition matrices with rank $r = 1$ and dynamically expand its rank individually during training to adapt to varying layer needs. Now, to facilitate adaptive growth, we address two key research questions: (1) How can to expand the rank **without perturbing the output**? (2) At what juncture should the expansion occur?

### 3.1 How to expand?

When *SeLoRA* expands itself by adding a new rank, the model's final prediction should remain constant. Hence, to prevent perturbations in the model's output when introducing a new rank, a straightforward approach is to force the product of the expanded rank to be 0. However, simple all-zero initialization poses a challenge as it hinders gradient flow through the expanded rank during initial back-propagation. Consequently, utilizing methods e.g., Fisher information to assess the expansion is not applicable. Therefore, we propose to initialize the expanded rank of matrix $B$ to be 0, and the expanded sub-matrix, $K$, of matrix $A$ to be randomly initialized. Hence, the expanded form of *SeLoRA* can be expressed as follows:

$$f(x) = xW_0 + x\begin{bmatrix} A & K \end{bmatrix}\begin{bmatrix} B \\ 0 \end{bmatrix} + b_0, \tag{2}$$

where $K \in \mathbb{R}^{d_{in} \times 1}$ is a vector initialized with Kaiming uniform initialization. Now, the expanded *SeLoRA* maintains the desired output while allowing the gradient to propagate through $A$.

### 3.2 When to expand?

To determine when an expansion could enhance the model, a crucial criterion is evaluating the potential improvement introduced by the rank addition. In assessing the viability of an expanded *SeLoRA* without excessive training, we employ Fisher information to measure the information conveyed by *SeLoRA* weights from the datasets. The conceptualisation of using Fisher information for expansion decision, stems from prior works on model selection procedures (using Fisher information), such as the Akaike information criterion, the Bayesian information criterion, and later work on low-rank approximation on neural network's weights Akaike (1974); Raftery (1995); Hsu et al. (2022). Here we utilized the empirically estimated Fisher information, introduced in Equation 3, as deriving an exact value is generally intractable due to the need for marginalization over the entire dataset.

$$\hat{I}_w = \frac{1}{|B|}\sum_{i=1}^{|B|}\left(\frac{\partial}{\partial w}\mathcal{L}(b_i; w)\right)^2, \tag{3}$$

where $|B|$ is the batch size, and $b_i$ is a sample in the batch. To quantify the information carried by a single *SeLoRA*, we introduce the Fisher information score (FI-Score), which sums over all Fisher information of the weight matrices $A, B$ in *SeLoRA*, as shown in Equation 4:

$$\text{FI-Score} = \sum_{i=1}^{d_{in}}\sum_{j=1}^{r}\hat{I}_{A_{i,j}} + \sum_{i=1}^{r}\sum_{j=1}^{d_{out}}\hat{I}_{B_{i,j}}. \tag{4}$$

Here, while the FI-Score is calculated by back-propagating gradients, it is important to note that the optimizer does not update the parameters during this calculation for the expanded *SeLoRA*. To further assess whether the expanded *SeLoRA* is superior to the previous unexpanded version, we introduce the Fisher information Ratio (FI-Ratio). The ratio between the Fisher information score of the expanded and original *SeLoRA* is calculated, as shown in Equation 5:

$$\text{FI-Ratio} = \frac{\text{FI-Score}_{orig}}{\text{FI-Score}_{exp}}. \tag{5}$$

Equation 5 provides a metric to evaluate the information gained when *SeLoRA* is expanded. This ratio is measured at each *SeLoRA* module, and it undergoes expansion when the FI-Ratio exceeds a desired threshold $\lambda$. An aggressive threshold, such as $\lambda = 1$, accepts any improvement to the model. Alternatively, a conservative threshold, such as $\lambda = 1.3$ or a larger value, could result in a more cautious expansion. Additionally, to ensure each expansion is beneficial, a hyperparameter $t$ is introduced. Testing of *SeLoRA*'s expansion is conducted at each $t$ training step, allowing the previous expanded rank to learn and converge before testing the new expansion. The detailed training procedure is shown in Algorithm 1.

---

**Algorithm 1** *SeLoRA* Training Procedure

---

**Initialization:** Initialize $W = W_0 + AB$ for each linear layer with $A$, $B$ at rank $r = 1$.
**Training:**
**While** $s <$ Total Steps:
    Forward pass $h = x(W_0 + AB) + b_0$.
    Update $A$ and $B$ with gradients $\nabla_A \mathcal{L}$ and $\nabla_B \mathcal{L}$.
    **If** $s \mod t = 0$:
        **For** each *SeLoRA* module:
            Compute original and expanded FI-Scores for $A$, $B$ and $A'$, $B'$.
            $A' = \begin{bmatrix} A & K \end{bmatrix}$, $B' = \begin{bmatrix} B^T & 0 \end{bmatrix}^T$.
            FI-Ratio $= \frac{\text{FI-Score}_{\text{orig}}}{\text{FI-Score}_{\text{exp}}}$.
            **IF** FI-Ratio $\geq \lambda$ **then** Update $A \leftarrow A'$, $B \leftarrow B'$.
            **EndIf**
        **EndFor**
    **EndIf**
    $s \leftarrow s + 1$.
**EndWhile**

---

### 3.3 ON CONVERGENCE OF RANK EXPANSION ALGORITHM

While *SeLoRA* can dynamically expand its rank and determine the timing of expansion, a question arises: Will *SeLoRA* continue to grow indefinitely without ever converging to an optimal rank? To address this concern, we demonstrate the existence of a theoretical upper-bound for the rank determined by the rank expansion algorithm. To analyze the behavior of the rank expansion, we let $c$ be the average fisher information across a *SeLoRA* module. Now, the FI-Ratio can be approximated as:

$$\text{FI-Ratio} = \frac{(\text{rank} + 1) \times (\text{in\_size} + \text{out\_size}) \times c_{expand}}{\text{rank} \times (\text{in\_size} + \text{out\_size}) \times c_{original}},$$

where rank is the current rank of *SeLoRA* at the layer being examined, and in\_size and out\_size are the input and output dimensions of the layer being examined, respectively.

When the rank of a *SeLoRA* is small, it tends to favor expansion due to the influence of the terms associated with rank. This behaviour is advantageous because, at lower ranks, the parameter count remains minimal, and the potential performance gains far outweigh the minor cost of adding additional parameters. As the rank grows, the FI-Ratio strikes a balance between (rank + 1) / rank term and any notable Fisher information figures in the expanded *SeLoRA* that significantly increase $c_{expand}$.

Once the rank reaches a sufficiently large value, the boost in FI-Ratio from the $(\text{rank} + 1)/\text{rank}$ ratio diminishes. As a result, further expansion will only occur if there is a substantial difference between the mean of expanded and original FI-Score, $c_{expand}$ and $c_{original}$. This mechanism naturally discourages unnecessary expansion at higher ranks, thereby preventing overfitting and ensuring parameter efficient fine tuning.

Hence, the FI-Ratio is modulated by both the rank and the significance of the update, ensuring a balance between expansion and avoiding overfitting. To further mitigate the risk of overfitting, a larger expansion threshold can be chosen.

### 3.4 DATASETS FOR EVALUATION

**IU X-RAY Dataset** Yang et al. (2022), collected by Indiana University, consists of 3,955 radiology reports paired with Frontal and Lateral chest X-ray images, each accompanied by text findings. For our experiments, we arbitrarily chose to exclusively focus on frontal projections, considering the substantial dissimilarities between frontal and lateral images. The dataset is partitioned into training, validation, and testing sets, with $80\%/16\%/4\%$ of the datasets, respectively.

Table 1: Quantitative results of IU X-RAY dataset. The best results are highlighted in pink.

| METHODS | FID ↓ | CLIP Score ↑ | FID 64 ↓ | FID 192 ↓ |
|---------|-------|--------------|----------|-----------|
| *LoRA* | $113.37 \pm 7.88$ | $27.11 \pm 0.18$ | $0.778 \pm 0.165$ | $5.993 \pm 1.878$ |
| *AdaLoRA* | $184.25 \pm 53.05$ | $26.90 \pm 0.33$ | $1.155 \pm 0.440$ | $12.288 \pm 5.186$ |
| *DyLoRA* | $116.03 \pm 10.29$ | $26.89 \pm 0.05$ | $1.018 \pm 0.484$ | $116.032 \pm 10.289$ |
| *SeLoRA* | **$113.04 \pm 16.79$** | **$27.26 \pm 0.22$** | **$0.536 \pm 0.332$** | **$5.004 \pm 3.211$** |

Table 2: Quantitative results of CXR dataset. The best results are highlighted in pink.

| METHODS | FID ↓ | CLIP Score ↑ | FID 64 ↓ | FID 192 ↓ |
|---------|-------|--------------|----------|-----------|
| *LoRA* | $461.80 \pm 14.10$ | $24.34 \pm 0.71$ | $29.002 \pm 7.485$ | $161.126 \pm 42.672$ |
| *AdaLoRA* | $489.24 \pm 14.28$ | $22.47 \pm 0.23$ | $45.448 \pm 3.499$ | $246.048 \pm 12.454$ |
| *DyLoRA* | $475.42 \pm 7.84$ | $22.93 \pm 0.27$ | $42.629 \pm 3.155$ | $245.815 \pm 14.100$ |
| *SeLoRA* | **$205.54 \pm 9.56$** | **$26.38 \pm 0.04$** | **$2.673 \pm 0.196$** | **$10.732 \pm 0.674$** |

**Montgomery County CXR Dataset** is relatively compact, consisting of 138 frontal chest X-rays (Demner-Fushman et al., 2015). Each entry in the dataset includes an X-ray image, along with details about the patient's sex, age, and medical findings. Among the 138 cases, 80 are labeled as 'normal,' with the remaining cases describing specific patient illnesses. The patient's sex, age, and findings are concatenated as prompt. The dataset is then divided into training and testing sets with a split ratio of $80\%/20\%$. Omitting a validation set ensures a sufficient number of samples for both the training and testing phases.

## 3.5 IMPLEMENTATION DETAILS

The experiments utilize the base Stable Diffusion model weights from 'runwayml/stable-diffusion-v1-5' in Hugging Face (Rombach et al., 2022). The Stable Diffusion model consists of three components: variational autoencoder (VAE), denoising U-Net, and the text encoder. When applying *LoRA* and its variants to Stable Diffusion, we injected them into every linear layer of the denoising U-Net and the text encoder. Injection into the VAE was omitted, as previous work has shown that fine-tuning the VAE component of the Stable Diffusion model does not yield significant performance improvements on medical images (Chambon et al., 2022b). Additionally, to accommodate the text encoder of stable diffusion, any text finding longer than 76 tokens is truncated, as outlined in the *CLIP* approach (Radford et al., 2021).

In our experiments, we compared *LoRA*, *DyLoRA*, and *AdaLoRA* to our proposed methods. We fine-tuned the model using a mean squared error (MSE) loss, evaluating it on the validation set after each epoch, and kept the best-performing model for testing. The models were fine-tuned for 10 epochs on the IU X-RAY dataset and 100 epochs on the Montgomery County CXR dataset, due to its smaller size. For our method, we selected a threshold of $\lambda = 1.1$ and $t = 40$ training steps. Although these parameter choices are somewhat arbitrary, a relatively large $t$ is preferred to allow *SeLoRA* to fully converge before evaluating expansion. For *LoRA* and *DyLoRA*, we set the rank to $r = 4$, while *AdaLoRA* was initialized with $r = 6$ and reduced to the target rank of $r = 4$. The choice of $r = 4$ ensures a comparable number of trainable parameters to our trained *SeLoRA*, enabling a fair comparison.

## 3.6 EVALUATION METRIC

We assessed the synthetic images based on both the fidelity and their alignment with the provided prompts. To assess the alignment with prompt, CLIP score is used, which measures text-to-image similarity (Radford et al., 2021).

To measure fidelity, we've used Fréchet Inception Distance (FID), which calculates the similarity between the distribution of synthetic datasets and the distribution of the original datasets (Heusel et al., 2017).

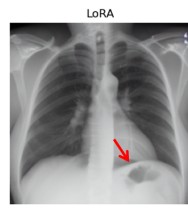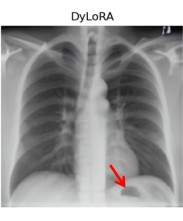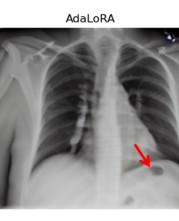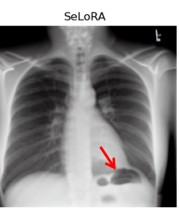

Figure 2: Qualitative comparison results obtained on the IU X-RAY data, generated by fine-tuning stable diffusion models injected with various LoRA variants. Prompt used: ``Heart size and vascularity normal. These contour normal. Lungs clear. No pleural effusions or pneumothoraces.'' More sample results are presented in the Appendix.

The original FID score is computed using the output of the final layer of a trained Inception v3 model, which produces a 2048-dimensional vector. To further evaluate the model's quality and ensure consistent results, we also examined the FID score using lower-dimensional latent representations of the synthetic images obtained from intermediate layers of the Inception v3 model. Specifically, we chose layers with latent vector dimensions of 64 and 192.

All metric results are averaged across three random seeds and summarized in Table 1 and 2 for IU X-RAY and Montgomery County CXR Datasets, respectively.

## 4 RESULTS

Our results clearly highlight the superior performance of the proposed method, *SeLoRA*, in terms of both fidelity and alignment with the prompt (text condition). In Tables 1 and 2, *SeLoRA* consistently outperforms all other methods across evaluation metrics for these relatively small datasets. This demonstrates *SeLoRA*'s potential and suggests it is a promising candidate for further exploration and application in various medical image synthesis tasks, especially given the inherent scarcity of medical datasets Papanastasiou et al. (2023).

Notably, *SeLoRA* achieves superior performance on the IU X-RAY dataset while utilizing only small fractions of the trainable parameters, averaging $0.121\%$ of total parameter for the text encoder and $0.368\%$ of total parameter for U-net part. In contrast, other *LoRA* methods with rank $r = 4$ use nearly double the trainable parameters ($0.216\%$ and $0.803\%$ of total parameter, respectively) but still achieve lower scores. This efficiency highlights *SeLoRA*'s effectiveness in parameter efficient training, particularly in memory-constrained environments.

To verify the robustness of *SeLoRA*, we also assessed FID scores using lower-dimensional image representations obtained from intermediate layers of the Inception v3 model. The results, shown in the rightmost two columns of Tables 1 and 2, align with previous findings, reaffirming that *SeLoRA* consistently outperforms other methods. Additionally, we observed a noticeable improvement in FID scores for the IU X-Ray dataset when evaluated with lower dimension. While the original FID scores were nearly tied with *LoRA* for the IU X-Ray dataset, our method demonstrates clear superiority when assessed across varying dimensions of FID score. This further highlights *SeLoRA*'s robustness and its suitability for medical image synthesis, especially in scenarios where traditional methods may struggle to maintain performance.

**Qualitative Results** Furthermore, synthetic images generated by a model trained on the IU X-RAY dataset are displayed in Figure 2. Notice that, although all synthetic images achieved similar quality, in comparison, *SeLoRA* is the only method that captures the distinct pathology (represented as a black circle) in the lower part of the right lung, closely resembling the original image. More sample images are shown in appendix Figure 7 and 9.

### 4.1 RANK ALLOCATION ANALYSIS

Figure 3 and 4 illustrates the final rank results of *SeLoRA* across stable diffusion model layers after training on the IU X-RAY dataset. For both figure, each cell within the figure represents a layer in

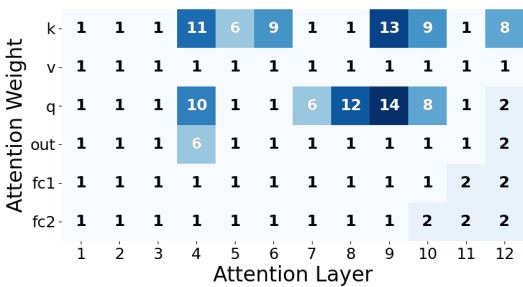

Figure 3: The final rank of *SeLoRA* in text encoder fine-tuned on stable diffusion with IU X-RAY Dataset. The x-axis labels correspond to the layer indices, while the y-axis labels denote different attention weights within the model. Each element represents the rank assigned to the corresponding attention weight in a specific layer. Note that *SeLoRA* places more rank on crucial layers.

Figure 4: The final rank of *SeLoRA* in Unet encoder fine-tuned on stable diffusion with IU X-RAY Dataset. The x-axis labels correspond to the layer indices, while the y-axis labels denote different attention weights within the model. Each element represents the rank assigned to the corresponding attention weight in a specific layer. Note that *SeLoRA* places more rank on crucial layers.

Table 3: Ablation study on performance of *SeLoRA* trained with different, threshold $\lambda \in \{1, 1.1, 1.3\}$, on IU X-RAY Dataset.

| Threshold | FID ↓ | CLIP Score ↑ |
|---|---|---|
| 1 | 76.442 | 27.265 |
| 1.1 | 113.042 | 27.256 |
| 1.3 | 122.752 | 26.835 |

the model, and the displayed number indicates the final learnt *SeLoRA* rank. The layers are arranged from input to output, top to bottom and left to right.

In figure 3 , *q, k,* and *v* represent the query, key, and value sections of the attention weights, respectively. *out*, *fc1*, and *fc2* denote the output, first fully connected layer, and second fully connected layer of the text encoder part. Similarly, for figure 4, *q, k,* and *v* represent the query, key, and value sections of the attention weights. *attn1* and *attn2* refer to the first and second attention layers, where the first is self-attention and the second is cross-attention between text and image embedding.

In the text encoder part, large ranks for *SeLoRA* lie at the *q* and *k* parts of the attention weights. For the U-net part, large ranks are allocated to the *q* and *k* parts of the second attention layer, namely the cross-attention layer. The rank allocation aligns with the intuition that weight updates would change most dramatically at locations where the latent representations of text and image intersect (where conditioning is more apparent). Hence, it validates our hypothesis that our proposed expansion method allows *SeLoRA* to focus and place more rank on crucial layers.

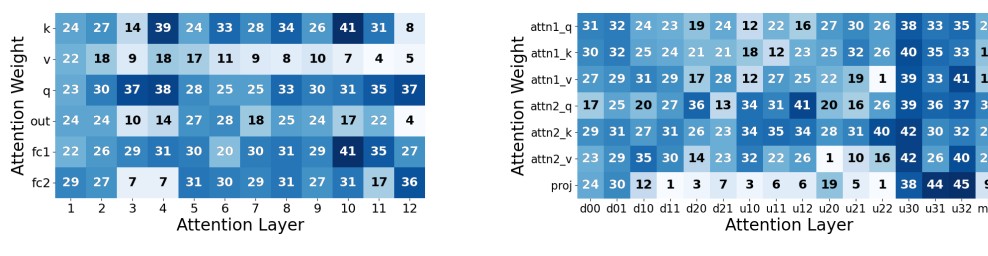

(a) Rank of *SeLoRA* in Text Encoder.

(b) Rank of *SeLoRA* in Denoising U-Net.

Figure 5: The final rank of each *SeLoRA* fine-tuned on stable diffusion with threshold of $\lambda = 1$ for IU X-RAY dataset. The x-axis represents the layer index, and the y-axis indicates the corresponding attention's weight name. *SeLoRA* focuses and places more rank on crucial layers.

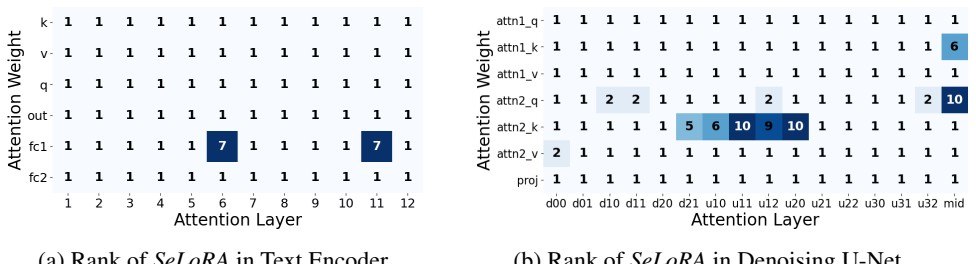

(a) Rank of *SeLoRA* in Text Encoder.

(b) Rank of *SeLoRA* in Denoising U-Net.

Figure 6: The final rank of each *SeLoRA* fine-tuned on stable diffusion with threshold of $\lambda = 1.3$ for IU X-RAY dataset. The x-axis represents the layer index, and the y-axis indicates the corresponding attention's weight name. *SeLoRA* focuses and places more rank on crucial layers.

## 4.2 ABLATION STUDY - THE IMPACT OF $\lambda$

Further results obtained by experimenting with different $\lambda$ value are displayed in Table 3. These results indicate that a smaller $\lambda$ leads to a more aggressive expansion of all layers in *SeLoRA*, resulting in improved synthesis quality, albeit at the cost of a more expensive training process.

The final learned rank results of *SeLoRA* for $\lambda = 1$ and $1.3$ are shown in Figures 5 and 6, respectively. The critical layers that *SeLoRA* emphasizes remain consistent with our observations in Section 4.1. Note that although the ranks in Figure 6 are spread out and appear quite large, when averaged across different layers, the $q$ and $k$ layers in the text encoder and the $attn2\_q$ and $attn2\_k$ layers still have the largest ranks. This further confirms the consistency of our method, aligning with our analysis in Section 4.1.

The significant discrepancies in the ranks assigned to each layer of the model, as shown in Figures 3, 4, 5, and 6, are expected. This phenomenon arises from the substantial differences in expansion thresholds used in the experiments. For Figure 5, the expansion ratio was set to $\lambda = 1$ for ablation, which almost always allows for rank expansion, naturally leading to higher ranks across the model's layers. In contrast, the experiment for Figure 6 used a larger threshold of $\lambda = 1.3$, which suppresses rank expansion across layers.

## 5 CONCLUSION

This work introduces a novel parameter-efficient method named *SeLoRA*, designed to effectively fine-tune stable diffusion models for generating X-ray images based on text (radiology) prompts. Our method enables progressive expansion in the rank of *LoRA*, enabling more precise image synthesis with minimal added rank. Through exploratory analysis, we demonstrate that the proposed FI-Ratio is capable of effectively guiding *SeLoRA* to expand its rank and allocate more rank to crucial layers. We believe that *SeLoRA*, when combined with stable diffusion, can be easily employed to adapt to various medical datasets containing text-image pairs, and potentially being applicable for clinical text synthesis. Moreover, given the increasing work on segmentation and detection in 3D

(tomographic, e.g., magnetic resonance imaging, computed tomography, and other) medical imaging data and the emergence of prompt-to-3D models, in our future work, we aspire to explore *SeLoRA* adaptations on fine-tuning prompt-to-3D models for 3D medical image synthesis.

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

# A APPENDIX

## A.1 ADDITIONAL QUALITATIVE RESULTS

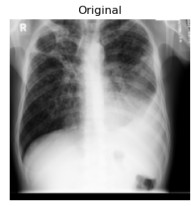 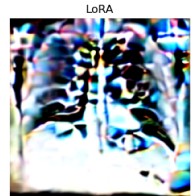 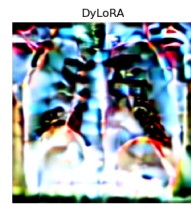 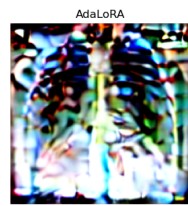 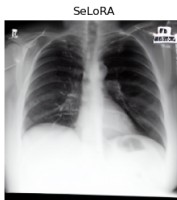

Figure 7: Qualitative comparison between real and synthetic images on the Montgomery County CXR data, generated by fine-tuning frozen stable diffusion models injected with LoRA variants.

**Prompt**: *" age:32, gender:Male, findings:extensive infiltrates bilaterally with large cavity in RUL and a moderate pleural effusion on the left. AFB smears and RNA probes pos for MTB. Active TB, cavitary."*

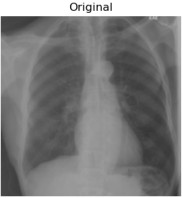 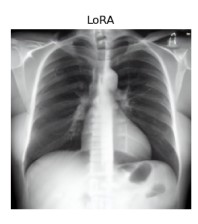 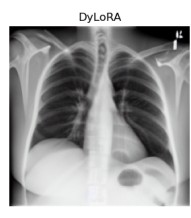 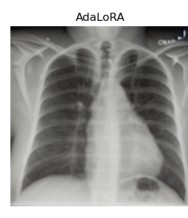 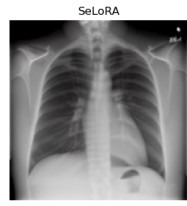

Figure 8: Qualitative comparison between real and synthetic images on the IU-Xray data, generated by fine-tuning frozen stable diffusion models injected with LoRA variants.

**Prompt**: *" The heart and lungs have XXXX XXXX in the interval. Both lungs are clear and expanded. Heart and mediastinum normal."*

# B TRAINING PROTOCOL

## B.1 DATA ENGINEERING

To accommodate varying image sizes, we first rescaled the images so that the shortest side measured 224 pixels. Center cropping was then applied to standardize all images to a size of $3 \times 224 \times 224$ (3 channels, 224 pixels height, and 224 pixels width).

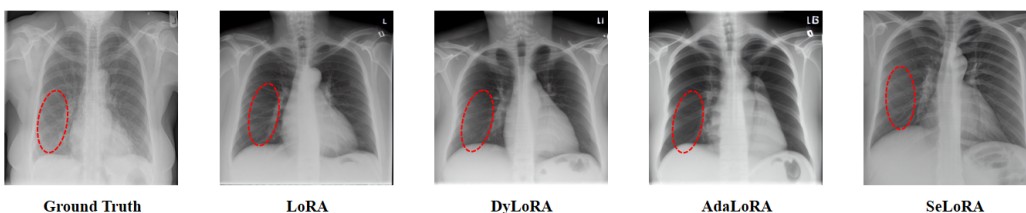

Figure 9: Qualitative comparison between real and synthetic images on the IU X-Ray dataset at $384 \times 384$ resolution, generated by fine-tuning frozen stable diffusion models injected with LoRA variants. Our method shows the most consistent match with the findings of "minimal patchy bibasilar airspace opacities" at the circled area.

**Prompt**: *"Minimal patchy bibasilar airspace opacities, XXXX atelectasis or evolving pneumonia. The heart pulmonary XXXX appear normal. Is minimal blunting of the pleural spaces, XXXX XXXX effusions."*



Figure 10: Qualitative comparison between real and synthetic images on the PatchGastricADC22 dataset, generated by fine-tuning frozen stable diffusion models injected with LoRA variants. Our method shows the most consistent match with the findings of "cells are highly columnar" at the annotated area.

**Prompt**: *"On the superficial epithelium, tumor tissue with densely growing medium to large, round tubules is observed. Tumor cells are highly columnar, with nuclei aligned basolaterally and polarized. Well differentiated tubular adenocarcinoma"*

## B.2    TRAINING DETAILS

Table 4: Training Configuration

| CONFIGURATION | VALUE |
| --- | --- |
| Optimizer | Adam |
| Base Learning Rate | $1 \times 10^{-4}$ |
| Weight Decay | 0 |
| Optimizer Momentum | $\beta_1 = 0.9, \beta_2 = 0.999$ |
| Batch Size | 8 |
| Training Epochs | 10 (IU X-RAY), 100 (CXR) |
| $\lambda$ | 1.1 |
| $t$ | 40 |

Table 5: Quantitative performance results on the IU X-Ray dataset at $384 \times 384$ resolution. The best results are highlighted in pink.

| METHOD | FID ↓ | FID 64 ↓ | FID 192 ↓ |
|--------|-------|----------|-----------|
| *LoRA* | 69.936 | 0.445 | 3.068 |
| *AdaLoRA* | 153.804 | 0.520 | 5.681 |
| *DyLoRA* | 89.765 | 0.580 | 4.982 |
| *SeLoRA* | **65.780** | **0.344** | **2.412** |

## C  EVALUATION AT HIGHER RESOLUTIONS AND ACROSS MODALITIES

### C.1  EXPERIMENT ON IU X-RAY DATASET AT $384 \times 384$ RESOLUTION

To further validate our approach, we trained and tested the method on $384 \times 384$ resolution images from the IU X-Ray dataset. For robust evaluation, the dataset was repartitioned into a 70%-10%-20% split for training, validation, and testing, respectively.

All experimental setup parameters, except for the batch size, remained consistent with those outlined in Section 3.5. The batch size was reduced to 6 to address GPU memory constraints. Frechet Inception Distance (FID) scores were computed across various embedding dimensions: original, 64, and 192. The results are summarized in Table 5.

As shown in Table 5, *SeLoRA* consistently achieves the lowest FID across all configurations, outperforming other methods significantly. These results demonstrate the robustness and superior performance of *SeLoRA* when evaluated at higher resolutions.

### C.2  ADDITIONAL EXPERIMENT ON PATCHGASTRICADC22 DATASET

To further strengthen our evaluation, we conducted additional experiments on a different modality. Specifically, we experimented on the **PatchGastricADC22** dataset (Tsuneki & Kanavati, 2022), a dataset comprising 262,777 image patches derived from 991 H&E-stained gastric slides. This dataset features adenocarcinoma subtypes and corresponding captions extracted from medical reports, representing a completely different modality from X-ray images. This experiment highlights the robustness and versatility of our method.

For our experimental setup, we used $384 \times 384$ image patches. Due to the class imbalance in the dataset, we randomly selected five subclasses: (1) Moderately differentiated tubular adenocarcinoma, (2) Well-differentiated tubular adenocarcinoma, (3) Moderately to poorly differentiated adenocarcinoma, (4) Poorly differentiated adenocarcinoma, solid type, and (5) Poorly differentiated adenocarcinoma, non-solid type.

From each subclass, we sampled 1,000 image-text pairs, creating a balanced subset. This subset was split into training, validation, and testing sets with a 70%-10%-20% partition. We evaluated the FID across various embedding dimensions (original, 64, and 192). The experimental setup and hyper-parameters, except for the number of images per batch, were consistent with Section 3.5. The number of images per batch was reduced to 6 to accommodate the GPU memory limitations. The results are shown in Table 6.

On the original dimension, we can notice that *SeLoRA* ranks second, behind *LoRA* by only 0.5, while significantly outperforming *AdaLoRA* and *DyLoRA*. On dimensions 64 and 192, *SeLoRA* surpasses *LoRA* but slightly under performs compared to *DyLoRA*. However, given *DyLoRA*'s worst performance in the original dimension, we believe *SeLoRA* demonstrates greater stability across embedding dimensions.

Table 6: Quantitative performance results on the PatchGastricADC22 dataset at $384 \times 384$ resolution. The best results are highlighted in pink.

| METHODS | FID ↓ | FID 64 ↓ | FID 192 ↓ |
|---|---|---|---|
| LoRA | **75.006** | 6.078 | 16.762 |
| AdaLoRA | 80.106 | 9.878 | 37.627 |
| DyLoRA | 93.083 | **5.002** | **14.376** |
| SeLoRA | 75.586 | 6.001 | 16.487 |

Table 7: The average training GPU memory usage recorded during experiments on each dataset at $384 \times 384$ resolution with a batch size of 6. The best results are highlighted in pink.

| METHODS | PatchGastricADC22 | IU X-Ray |
|---|---|---|
| LoRA | 21.591 | 20.508 |
| AdaLoRA | 20.908 | 19.644 |
| DyLoRA | 20.921 | **18.995** |
| SeLoRA | **20.751** | 19.179 |

## D TRAINING EFFICIENCY ANALYSIS

For both the **IU X-Ray** and **PatchGastricADC22** dataset trained on $384 \times 384$ resolution. We have recorded the GPU memory usage and training time for training efficiency analysis.

### D.1 GPU MEMORY USAGE DURING TRAINING

During the training of the **IU X-Ray** and **PatchGastricADC22** datasets at $384 \times 384$ resolution with a batch size of 6, we recorded the GPU memory usage for all methods every 40 steps. The details are shown in Figures 11 and 12.

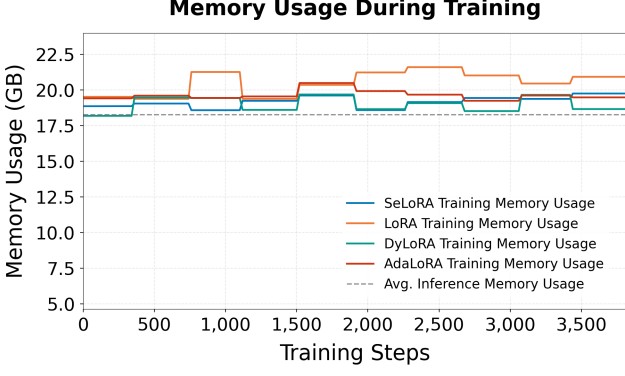

Figure 11: GPU memory usage recorded during experiments on the IU X-Ray dataset at $384 \times 384$ resolution with a batch size of 6. GPU memory usage was monitored and recorded every 40 steps. The upper and lower limits of the y-axis represent the maximum available GPU memory and the memory used for loading the model, respectively.

In both plots, the memory usage of *SeLoRA* is, in most cases, the lowest and was never the highest. To provide a clearer perspective, we computed the average memory usage across all timestamps and present the results in Table 7.

We observe that *SeLoRA* have the lowest average training GPU memory usage on the **PatchGastricADC22** dataset. This advantage may compensate for the slight 0.5-point disadvantage in FID when compared to *LoRA* method. On the **IU-XRay** dataset, although our method ranked second in

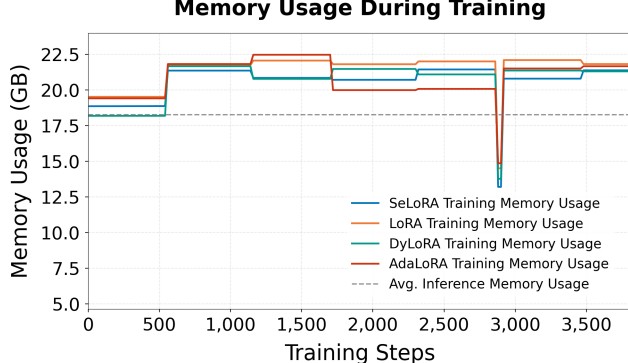

Figure 12: GPU memory usage recorded during experiments on the PatchGastricADC22 dataset at $384 \times 384$ resolution with a batch size of 6. GPU memory usage was monitored and recorded every 40 steps. The upper and lower limits of the y-axis represent the maximum GPU memory available and the memory used for loading the model, respectively.

Table 8: The average training time recorded during experiments on each dataset at $384 \times 384$ resolution with a batch size of 6.

| METHODS | PatchGastricADC22 | IU X-Ray |
|---------|-------------------|----------|
| *LoRA* | 2:40:15 | 1:44:43 |
| *AdaLoRA* | 2:37:42 | 1:49:44 |
| *DyLoRA* | 2:44:01 | **1:41:30** |
| *SeLoRA* | **2:36:56** | 1:47:01 |

GPU memory usage, behind *DyLoRA*, it significantly outperformed *DyLoRA* in terms of FID, with a 23-point difference. This indicates that, despite a modest increase in memory usage, *SeLoRA* delivers superior FID performance, thus offering a highly effective balance between memory efficiency and overall performance.

### D.2 TRAINING TIME

We also recorded the training times for both the **IU X-Ray** and **PatchGastricADC22** datasets, as shown in Table 8.

On PatchGastricADC22 dataset, we observed a similar pattern as in training memory GPU Usage, with *SeLoRA* having the lowest training time on PatchGastricADC22. For IU X-Ray dataset, *SeLoRA* ranked the third. *SeLoRA* exhibits a slightly longer training time compared to *LoRA* and *DyLoRA*; however, the difference is minimal and does not diminish its advantages in model performance as in 5. This additional training time arises from the rank increment testing step, which is performed every 40 training steps as specified in the experimental setup.

### D.2.1 ESTIMATED WORST-CASE TRAINING TIME FOR *SeLoRA*

To provide a clear estimate of the worst-case training time for *SeLoRA* relative to the standard *LoRA* method, we assume that *LoRA* operates with a fixed rank equivalent to the average final rank achieved by *SeLoRA*. Denoting the total training time for *LoRA* as $\texttt{LoRA\_time}$, and the interval between rank increment tests in *SeLoRA* as $t$ steps, the total training time for *SeLoRA* can be approximated as:

$$\texttt{SeLoRA\_Time} = \left(1 + \frac{1}{t}\right) \times \texttt{LoRA\_time}. \qquad (6)$$

Here, $t$ represents the frequency of rank testing, and the term $\frac{1}{t}$ reflects the proportionate overhead incurred by performing rank testing during training. This equation models the worst-case scenario, where every rank test incurs an additional step, slightly increasing the overall training time.

Notably, this estimate is conservative, as *SeLoRA* typically operates with lower ranks than *LoRA* during most of the training process, leading to reduced computational costs. As a result, the actual training time for *SeLoRA* is often below this calculated worst-case threshold.

In our experiments, the training time for *LoRA* was approximately $104.71$ minutes, with rank testing performed every $t = 40$ steps. Substituting these values into the formula, the estimated training time for *SeLoRA* is:

$$\texttt{SeLoRA\_Time} = \left(1 + \frac{1}{40}\right) \times 104.71 = 107.32 \,\text{minutes.} \tag{7}$$

This estimate closely matches the experimentally observed training time for *SeLoRA*, confirming the validity of the approximation. Importantly, the additional overhead of approximately $2.6$ minutes (or $2.5\%$) is minimal and predictable, making it a reasonable trade-off for the significant gains in adaptability and performance achieved by *SeLoRA*.

In summary, this analysis highlights that *SeLoRA* maintains computational efficiency while introducing dynamic rank adjustments, with only a slight increase in training time compared to *LoRA*.

# E    NOTE ON DATASET SIZES FOR *LoRA* FINE-TUNING

As a reference for the size of datasets typically used to fine-tune *LoRA* on Stable Diffusion, a prior ICLR paper on *LoRA* adaptation for Stable Diffusion utilized around 45–200 images per class, totaling approximately 1,706 images, as noted by Yeh et al. (2024). Previous studies have employed the MIMIC-CXR dataset for fine-tuning, which is significantly larger, with 40,000 images per class. We believe that fine-tuning on smaller datasets is a more appropriate and practical application of *LoRA*-based methods, particularly in scenarios like ours, as opposed to training on the full 377,000-image MIMIC-CXR dataset.

