# OpenReview forum: "SeLoRA: Self-Expanding Low-Rank Adaptation of Latent Diffusion Model for Medical Image Synthesis"
_ICLR.cc/2025/Conference — Submitted to ICLR 2025_

### Official Review · Reviewer_Pa5H · 2024-10-28

**Soundness:** 3
**Presentation:** 3
**Contribution:** 3
**Rating:** 6
**Confidence:** 3

**Summary:**

This paper presents Self-Expanding Low-Rank Adaptation (SeLoRA), akin to LoRA’s structure but distinguished by the dynamic growth of ranks guided by Fisher information during training.   This enables SeLoRA to flexibly adapt to the inherent characteristics of each layer, guaranteeing enhanced medical image synthesis quality while minimizing challenges related to rank adjustments.

**Strengths:**

* Originality: good originality, combining self-expanding and LoRA to make LoRA self-adapt to the different needs of layers.

* Quality: simple but effective method. Performance validated in the scope of medical image synthesis.

* Clarity: clear figures and method presentation.

* Significance: Data scarcity is very common in the medical AI field, and a strong LoRA variant is important to the community. IF the proposed method is effective and generalizable in the general scenes, it could be significant in efficient learning.

**Weaknesses:**

* The paper proves the proposed method is significantly outstanding in the medical image synthesis scene (IU X-RAY dataset and CXR dataset). Would it work well in other tasks, like perceptual tasks or other synthesis tasks?

* Time cost is an important metric for the self-expand algorithm. The paper lacks a discussion of the efficiency of the algorithm and the experimental comparison with other LoRA methods.

**Questions:**

Included in the Weaknesses part.

---

> ### Author Response · Authors · 2024-11-25
> **Rebuttal from Authors of Paper10521 to Reviewer Pa5H**
>
> We appreciate Reviewer Pa5H’s valuable and constructive feedback regarding the analysis of computational complexity and the limited modalities in our experiments. In response to these concerns, we have conducted new analyses and additional experiments. Below, we address each point raised in detail:
>
> **Including Other Modalities**
>
> To address this, we have included an additional experiment on pathology images - the PatchGastricADC22 dataset, demonstrating the robustness and versatility of our method. The results are shown in Table 6 and are also showcased below:
>
> | **METHODS**   | **FID ↓ (PatchGastricADC22)**            | **FID 64 ↓ (PatchGastricADC22)** | **FID 192 ↓ (PatchGastricADC22)**     |
> | ------------- | ---------------------------------------- | -------------------------------- | ------------------------------------- |
> | *LoRA*        | **75.006**                               | 6.078                            | 16.762                                |
> | *AdaLoRA*     | 80.106                                   | 9.878                            | 37.627                                |
> | *DyLoRA*      | 93.083                                   | **5.002**                        | **14.376**                            |
> | *SeLoRA*      | 75.586                                   | 6.001                            | 16.487                                |
>
> On the PatchGastricADC22 dataset, while our method ranked second, it is comparable to the best-performing method (*LoRA*) and only 0.5 points behind. Additionally, our method exhibited the **lowest training time and GPU memory usage** on PatchGastricADC22 as shown below, justifying the slight degradation in FID. This demonstrates a favorable trade-off between performance and training efficiency. Furthermore, we plan to validate our method on additional modalities, such as MRI, CT, or ultrasound datasets, in future work.
>
> **Training Efficiency Analysis**
>
> To facilitate a training efficiency analysis, we conducted experiments for *SeLoRA* and related methods using the IU X-ray dataset, as well as the PatchGastricADC22 dataset. During these experiments, we recorded the training time and GPU memory usage at intervals of every 40 training steps. The results are presented in Table 8 (training time) and Table 7, Figure 11, and Figure 12 (GPU memory usage). For ease of viewing, the training time and average GPU memory usage are summarized below:
>
> | **METHODS**   | **GPU Memory Usage (PatchGastricADC22)** | **GPU Memory Usage (IU X-Ray)** | **Training Time (PatchGastricADC22)** | **Training Time (IU X-Ray)** |
> |----------------|-----------------------------------------|----------------------------------|---------------------------------------|-------------------------------|
> | *LoRA*        | 21.591                                  | 20.508                           | 2:40:15                              | 1:44:43                      |
> | *AdaLoRA*     | 20.908                                  | 19.644                           | 2:37:42                              | 1:49:44                      |
> | *DyLoRA*      | 20.921                                  | **18.995**                       | 2:44:01                              | **1:41:30**                  |
> | *SeLoRA*      | **20.751**                              | 19.179                           | **2:36:56**                          | 1:47:01                      |
>
> For training time, our method achieved the **shortest training time** on the PatchGastricADC22 dataset and ranked third for the IU X-Ray dataset. However, on the IU X-ray dataset, our method was only 6 minutes slower than the fastest method (DyLoRA) — a negligible difference. When analyzing training time in conjunction with the FID evaluation in Tables 5 and 6, we observe a significant advantage. On the IU X-Ray dataset, our method outperforms the fastest method (DyLoRA) by **23 points in FID**. Meanwhile, on the PatchGastricADC22 dataset, where our method is the fastest, it incurs only a minor 0.5-point disadvantage in FID compared to the best-performing method (LoRA). This demonstrates that our method offers a highly advantageous trade-off between training time and performance.
>
> For GPU memory usage displayed in Figures 11 and 12, it is evident that our method never required the highest GPU memory among the compared methods and consistently had the lowest memory usage. In Table 7, which reports the average GPU memory usage, our method had the **lowest memory usage** on the PatchGastricADC22 dataset and ranked second on the IU X-Ray dataset. Similar to the training time results, although our method is slightly behind DyLoRA on the IU X-Ray dataset in GPU memory usage, the significant 23-point advantage in FID justifies this trade-off.
>
> Therefore, we believe our method strikes the best balance between performance and training efficiency.

---

> > ### Comment · Reviewer_Pa5H · 2024-11-26
> >
> > Thanks for the author's hard work. Additional experiments solved my worries about the time cost. I will not change the rating.

---

> > > ### Author Response · Authors · 2024-11-28
> > >
> > > Thank you very much for your kind response, especially your constructive advices which leads to significant improvement of this paper.

---

### Official Review · Reviewer_7CbA · 2024-11-03

**Soundness:** 3
**Presentation:** 3
**Contribution:** 3
**Rating:** 6
**Confidence:** 5

**Summary:**

The paper present SeLoRA, a Self-Expanding Low-Rank Adaptation module, that dynamically expands its ranking across layers during training. The proposed method increases the rank from 1 gradually. FI-Ratio and parameter \lambda were used to determine when to expanding the rank.  The experiment is performed with stable diffusion and X-ray dataset and synthesis the X-ray image with text prompt.

**Strengths:**

The paper introduces a parameter-efficient method to e fine-tune stable diffusion models for generating X-ray images based on text (radiology) prompts. And the proposed method can progressive expansion in the rank of LoRA. FI-Ratio is used to guiding SeLoRA to expand its rank. The rank of different layers was given. The experiment shows the result is promisingly.

**Weaknesses:**

1. Computational overhead. While SeLoRA reduces the number of trainable parameters, its dynamic rank expansion mechanism introduces additional computational complexity. Computing Fisher information increases the overhead, which may become significant for larger datasets or more complex models.
2. Limited dataset evaluation. Experiments were limited to two 2D X-ray datasets, with no evaluation of SeLoRA’s performance on other modalities, such as MRI or CT scans. Further validation on additional modalities would help confirm the generalizability of the method.
3. Visual result is limited. The visual result is not satisfactory, such as in Figure 8, the contrast and details are not good.

**Questions:**

1. Computational complexity analysis A comparison of training time, memory usage and FLOPs between SeLoRA, LoRA, and other variants is needed to quantify the computational trade-offs introduced by dynamic rank expansion.
2. Evaluation on a wider range of datasets Evaluating SeLoRA on larger datasets, such as MIMIC-CXR (containing approximately 377,000 images), would provide more insights into its scalability. Future work could also validate SeLoRA on MRI, CT, or ultrasound datasets, as testing on diverse datasets would better demonstrate its robustness and versatility.
3. Incorporating related work. The idea of dynamically adjusting the rank of the LoRA matrix in SeLoRA is conceptually similar to the recently proposed ALoRA (NAACL 2024). However, the two methods differ in implementation: ALoRA utilizes pruning and redistribution strategies, while SeLoRA relies on Fisher information as the adjustment criterion. Insights from ALoRA could provide valuable inspiration for future improvements of SeLoRA.
4. Evaluation with medical doctor may help verify the experiment results.
Other questions
1. Unconventional split of the IU X-Ray dataset. The 80:16:4 split results in a relatively small test set, which could compromise the robustness of the evaluation. A more conventional split (e.g., 80:10:10) might provide more reliable insights.
2. Small sample size in the Montgomery County CXR dataset. With only 138 samples, the Montgomery County dataset is too small for deep learning applications, which may impact the stability and generalizability of the model’s results

---

> ### Author Response · Authors · 2024-11-25
> **(1/2) Rebuttal from Authors of Paper10521 to Reviewer 7CbA**
>
> We appreciate Reviewer **7CbA**'s thorough feedback, particularly regarding the analysis of computational complexity and our evaluation. We have strengthened our work by conducting additional experiments and analyses to address these concerns. Our detailed responses follow:
>
> **Computational Complexity Analysis**
>
> To facilitate computational complexity analysis, we conducted experiments for _SeLoRA_ and related methods using the IU X-ray dataset, as well as an additional PatchGastricADC22 dataset. During these experiments, we recorded the training time and GPU memory usage at intervals of every 40 training steps. The results are presented in Table 8 (training time) and in Table 7, Figure 11, and Figure 12 (GPU memory usage). For ease of viewing, the training time and average GPU memory usage are summarized below:
>
> |**METHODS**|**GPU Memory Usage (PatchGastricADC22)**|**GPU Memory Usage (IU X-Ray)**|**Training Time (PatchGastricADC22)**|**Training Time (IU X-Ray)**|
> |---|---|---|---|---|
> |_LoRA_|21.591|20.508|2:40:15|1:44:43|
> |_AdaLoRA_|20.908|19.644|2:37:42|1:49:44|
> |_DyLoRA_|20.921|**18.995**|2:44:01|**1:41:30**|
> |_SeLoRA_|**20.751**|19.179|**2:36:56**|1:47:01|
>
> **Key results**:
>
> * *SeLoRA* achieves the best FID for both datasets.
> * *SeLoRA* consistently balances training time, memory usage, and performance.
> * The additional overhead of Fisher Information computation is minimal, making *SeLoRA* efficient even in resource-constrained settings.
>
> For the training time, our method has the **shortest training time** on the PatchGastricADC22 dataset and ranks third for the IU X-ray dataset. However, on the IU X-ray dataset, our method is only slightly behind the fastest method (_DyLoRA_) by just 6 minutes—a negligible difference. When analyzing training time in conjunction with the FID evaluation in Tables 5 and 6, we observe a significant advantage. On the IU X-ray dataset, our method outperforms the fastest method (_DyLoRA_) by **23 points in FID**. Meanwhile, on the PatchGastricADC22 dataset, where our method is the fastest, we incur only a minor 0.5-point disadvantage in FID compared to the best-performing method (_LoRA_). We believe this suggests that our method offers a highly advantageous trade-off between training time and performance.
>
> For GPU memory usage displayed in Figures 11 and 12, it is evident that our method never required the highest GPU memory among the compared methods and consistently used the lowest memory. In Table 7, which reports the average GPU memory usage, our method had the **lowest memory usage** on the PatchGastricADC22 dataset and ranked second on the IU X-ray dataset. Similar to the training time results, although our method is slightly behind _DyLoRA_ on the IU X-ray dataset in GPU memory usage, the significant 23-point advantage in FID makes this trade-off justifiable.
>
> Therefore, we believe our method **strikes the best balance** between performance and training efficiency.
>
> **Computation Overhead for Fisher Information**
>
> Regarding the computational overhead for calculating Fisher information, it can be efficiently computed by taking the gradient and performing element-wise squaring of each matrix element. Therefore, the primary computational overhead comes from computing the model's gradient through backpropagation, while the additional overhead of squaring and summing the elements is negligible. If we estimate the additional computational complexity introduced by calculating Fisher information, it will be around $\frac{1}{t}$, where $t$ refers to the rank testing interval. In our case, with $t = 40$, this translates to an additional **2.5\%** overhead in terms of computational cost. Given this, we believe that the computation of Fisher information introduces a modest and manageable overhead, without significantly impacting the overall training efficiency.

---

> ### Author Response · Authors · 2024-11-25
> **(2/2) Rebuttal from Authors of Paper10521 to Reviewer 7CbA**
>
> **Evaluation of New Modality and Larger Datasets**
>
> To address this, we have included a new experiment on pathology images using the PatchGastricADC22 dataset to demonstrate the robustness and versatility of our method. The results are shown in Table 6. On the PatchGastricADC22 dataset, while our method ranked second, it was only 0.5 points behind the best-performing method (*LoRA*). Additionally, our method exhibited the lowest training time and GPU memory usage on PatchGastricADC22, justifying the slight increase in FID. This demonstrates a favorable trade-off between performance and training efficiency. Furthermore, we plan to validate our method on additional modalities, such as MRI, CT, or ultrasound datasets, in future work.
>
> Evaluating *SeLoRA* on larger datasets, such as MIMIC-CXR (which contains approximately 377,000 images), would provide further insights into its scalability. However, the primary challenge that medical image synthesis aims to address is the scarcity of labeled medical images. This is precisely why fine-tuning, rather than training from scratch, becomes crucial in such contexts. If a dataset of the scale of 377,000 images (comparable to ImageNet-1K) were available, there would be less incentive to generate additional synthetic data. In such cases, the focus would shift towards directly training models on the original dataset to leverage the abundant real-world data.
>
> As a reference for the size of datasets typically used to fine-tune *LoRA* on Stable Diffusion (SD), **a prior ICLR paper on *LoRA* adaptation for SD utilized around 45–200 images per class, totaling approximately 1,706 images**, as noted in Yeh et al., 2024. In contrast, MIMIC-CXR is significantly larger, with 40,000 images per class. We believe that fine-tuning on smaller datasets is a more appropriate and practical use case for *LoRA*-based methods, particularly in scenarios like ours, compared to training on the full 377,000-image MIMIC-CXR dataset.
>
> **Incorporating Related Work**
>
> We acknowledge that the idea of dynamically adjusting the rank of the *LoRA* matrix in *SeLoRA* shares conceptual similarities with the recently proposed *ALoRA* which has been tested language models, and we have included it in our related work section. Indeed, *ALoRA* could provide valuable inspiration. This caused our immediate interests in the two types of philosophy behind *ALoRA* and *SeLoRA*. In future work, we plan to explore the integration of pruning mechanisms alongside our Fisher information-guided rank adjustment to further enhance *SeLoRA’s* flexibility and efficiency. It would be especially interesting to compare *ALoRA's* efficiency, where it reach its peak of GPU memory usage at the beginning of training, aganist *SeLoRA*, where the the maximum of GPU memory usage arrived at the end.
>
> **Medical Doctor Evaluation**
>
> We have engaged medical doctors for evaluations that are currently underway. However, we do not anticipate completing these evaluations before the rebuttal deadline. We plan to include these expert assessments in future work to thoroughly evaluate the model's ability to generate clinically relevant images based on disease or abnormal findings. In the meantime, to address this concern partially, we have conducted additional experiments on the IU X-ray dataset with higher resolution, as well as on another modality using the PatchGastricADC22 dataset. These experiments demonstrate the robustness of our model.
>
> **Data Split of the IU X-Ray Dataset**
>
> To address this concern, we have repartitioned the IU X-Ray dataset into a 70%-10%-20% split for training, validation, and testing, respectively, to ensure a more robust evaluation with a larger test set. The results, presented in Table 5 in the updated manuscript, show that our method continues to achieve the best FID among all *LoRA*-based methods.

---

> > ### Comment · Reviewer_7CbA · 2024-11-28
> >
> > Thanks for the author's hard work. Additional experiments solved my questions about the time cost. I think I will not change the rating, even I think medical doctor evaluation is necessary.

---

> > > ### Author Response · Authors · 2024-11-28
> > >
> > > Thank you very much for constructive and detailed comments, which led to significant improvement of this paper!

---

### Official Review · Reviewer_Ko6A · 2024-11-03

**Soundness:** 2
**Presentation:** 3
**Contribution:** 2
**Rating:** 5
**Confidence:** 4

**Summary:**

This paper proposes a new parameter-efficient fine-tuning method SeLoRA (Self-Expanding Low-Rank Adaptation) for adapting Stable Diffusion to generate chest x-ray images. The main contribution of the paper lies in dynamically expanding the rank of LoRA during the training process, allowing it to adapt the rank according to the importance of different layers and thereby improving the quality of the synthesized images. The novelty of this approach is in using Fisher Information to guide the rank expansion, avoiding the limitations of the traditional LoRA method, which uses a "uniform rank," especially when dealing with models (like Stable Diffusion) that have diverse weight matrix shapes. The paper demonstrates the effectiveness of SeLoRA on two chest x-ray datasets and provides detailed comparative experiments with other LoRA variants.

**Strengths:**

1.
The papers proposes SeLoRA, a dynamic rank-expanding method using Fisher Information to guide rank expansion during fine-tuning large models with LoRA. This is novel and more applicable to models with diverse weight matrix shapes.
2.
Experimental results demonstrate the effectiveness of the proposed method.
3.
The paper is well-organized.

**Weaknesses:**

1.
The contribution is vague. As a paper focusing on adaptive parameter efficient fine-tuning methods, the paper utilizes LoRA to adapt Stable Diffusion for Chest X-ray synthesis, limiting its technical contribution. For a paper dedicated to adapting foundation model like Stable Diffusion for Chest X-ray (medical image) synthesis, the exploration is also limited and does not compare with previous work (e.g. Chambon et al., 2022a;b). Through visual comparison with image displays in Chambon et al., 2022a;b, the proposed method seems at a disadvantage.
2.
Experiments are conducted on relatively small datasets. Large image-report paired Chest X-ray datasets exist (e.g. MIMIC-CXR). Is this because of the heavy burden of large model like Stable Diffusion, or may be also related to the proposed method? Can the authors provide training time comparisons between SeLoRA and the compared methods? Also, as the test set of IU X-RAY and Montgomery County CXR dataset has only contains 100~200 images, the validation of the effectiveness of the method is weak.
3.
Evaluation and explanation are insufficient. Using a CLIP model trained purely on natural images and a maximum text token length of 76 to compute CLIP-score may not faithfully reflect how good the text-image alignment is for Chest x-ray images.
The training/validation/testing split is strange, could the author explain why the test set only contain 4% of the data? Are Table 1 values computed with valid data or test data? This is unclear. The paper also lacks in-depth discussion of the distribution of final rank (Figure 4,5,6) and why other LoRA methods fail on Montgomery county CXR data (Figure7). How accurate can the model generate an image given the disease or abnormal findings in the text prompt? This may be revealed using pretrained Chest X-ray classification models or manually inspect a small subset of generated results.

Other non-important issues:
4.
the paper title is about "medical image syntheis”; but it only focuses on chest x-ray image.
5.
The formula derivation in section3.3 is unclear.
6.
The Stable Diffusion model is trained to generate a resolution of 512x512, using a resolution of 224x224 may limit the performance.

**Questions:**

Please answer the questions mentioned in the above 'weakness' section.

---

> ### Author Response · Authors · 2024-11-25
> **(1/2) Rebuttal from Authors of Paper10521 to Reviewer Ko6A**
>
> We thank Reviewer **Ko6A** for the valuable and constructive feedback on the data size, limited modality, and the suggestion to explore higher image resolution. In response, we have conducted additional analyses and experiments to address these concerns. Below, we provide a point-by-point response:
>
> **Visual Quality Compared to Chambon et al. (2022a;b)**
>
> We acknowledge the difference in visual quality. Chambon et al. fine-tuned their model with 175k images using 64 A100 GPUs, whereas our method was fine-tuned with only 2k images on a single T4 GPU. Despite these disparities, *SeLoRA* stands out by prioritizing efficiency and adaptability in resource-constrained environments and low-data regimes, setting it apart from prior work. Additionally, we have provided further results at a higher resolution (384×384), highlighting improved visual quality and its potential for broader evaluations.
>
> **Size of Datasets**
>
> Evaluating *SeLoRA* on larger datasets, such as MIMIC-CXR, which contains approximately 377,000 images, could offer deeper insights into its scalability. However, the core challenge in medical image synthesis is **the scarcity of labeled medical images**. This underscores the importance of **fine-tuning** over training from scratch in such scenarios. If a large dataset like MIMIC-CXR (comparable to ImageNet-1K) were available, the need for generating synthetic data would diminish. In such cases, the focus would shift to directly training models on the original dataset to leverage abundant real-world data.
>
> As a reference, prior ICLR papers on *LoRA* adaptation for Stable Diffusion (SD), such as Yeh et al. (2024), utilized datasets containing approximately 45–200 images per class, totaling around 1,706 images. In contrast, MIMIC-CXR is significantly larger, with approximately 40,000 images per class. We believe fine-tuning on smaller datasets is a more appropriate and practical use of *LoRA*-based methods, particularly in scenarios like ours, compared to training on the full 377,000-image MIMIC-CXR dataset.
>
> To address concerns about small test sets, we conducted experiments using a repartitioned dataset with a 70%-10%-20% split for training, validation, and testing. The results, presented in Tables 5 and 6, show that our methods achieve the best FID for all dimensions on the IU X-Ray dataset.
>
> **Clarifications on Minors**
>
> **Dataset split**. We have conducted an experiment on a repartitioned dataset with a 70%-10%-20% split for training, validation, and testing.
>
> **Table 1**  was computed using the test dataset.
>
> ***SeLoRA*'s advantage.** The Montgomery County CXR dataset contains only about 100 training samples, compared to the larger IU X-Ray dataset used by other methods. In this low-data regime, most *LoRA*-based methods failed to learn meaningful representations and instead produced noisy or unrealistic lung images. This limitation stems from their uniform-rank or pruning designs, which hinder effective learning with small datasets.
>
> In contrast, our method dynamically allocates rank to the most important layers, guided by Fisher information. By adapting rank placement, *SeLoRA* captures the critical features necessary for generating high-quality images, even with limited data. This capability highlights *SeLoRA*'s distinct advantage in low-data settings, a strength that is especially vital in the medical imaging domain.
>
>
> **More evaluation.** Since there are no readily available chest image classification models specifically trained on the IU X-Ray dataset, we have engaged medical experts for evaluations. However, we do not anticipate having sufficient time to complete these evaluations before the rebuttal deadline. We plan to include these expert assessments in future work to provide a more thorough evaluation of the model’s ability to generate clinically relevant images. In the meantime, to partially address this concern, we conducted additional experiments on IU X-ray with **higher resolution** and on a new pathology dataset as below.

---

> ### Author Response · Authors · 2024-11-25
> **(2/2) Rebuttal from Authors of Paper10521 to Reviewer Ko6A**
>
> **Additional Experiments on New Dataset and Higher Resolution**
>
> **New dataset.** We acknowledge the reviewer’s observation regarding the focus of our work on chest X-ray images, given the title’s broader scope of "medical image synthesis". To address this, we have included an additional experiment on the PatchGastricADC22 dataset, representing a different modality and highlighting the robustness and versatility of our method. The results are shown in Table 6 and summarized below:
>
>
> | **METHODS**   | **FID ↓ (PatchGastricADC22)** | **FID 64 ↓ (PatchGastricADC22)** | **FID 192 ↓ (PatchGastricADC22)** |
> |----------------|------------------------------|----------------------------------|-----------------------------------|
> | *LoRA*        | **75.006**                   | 6.078                            | 16.762                            |
> | *AdaLoRA*     | 80.106                       | 9.878                            | 37.627                            |
> | *DyLoRA*      | 93.083                       | **5.002**                        | **14.376**                        |
> | *SeLoRA*      | 75.586                       | 6.001                            | 16.487                            |
>
> On the PatchGastricADC22 dataset, while our method ranked second, it was only 0.5 points behind the best-performing method (LoRA). Additionally, our method exhibited the lowest training time and GPU memory usage on PatchGastricADC22, justifying the slight increase in FID. This provides a good trade-off between performance and training efficiency.
>
> **Higher resolution.** We have conducted additional experiments on the PatchGastricADC22 and IU X-Ray datasets with a higher resolution of 384x384. The FID scores are reported in Tables 5 and 6. For convenience, we summarize the results below:
>
> | **METHODS**   | **FID ↓ (PatchGastricADC22)** | **FID 64 ↓ (PatchGastricADC22)** | **FID 192 ↓ (PatchGastricADC22)** | **FID ↓ (IU X-Ray)** | **FID 64 ↓ (IU X-Ray)** | **FID 192 ↓ (IU X-Ray)** |
> |----------------|------------------------------|----------------------------------|-----------------------------------|-----------------------|-------------------------|--------------------------|
> | *LoRA*        | **75.006**                   | 6.078                            | 16.762                            | 69.936                | 0.445                   | 3.068                   |
> | *AdaLoRA*     | 80.106                       | 9.878                            | 37.627                            | 153.804               | 0.520                   | 5.681                   |
> | *DyLoRA*      | 93.083                       | **5.002**                        | **14.376**                        | 89.765                | 0.580                   | 4.982                   |
> | *SeLoRA*      | 75.586                       | 6.001                            | 16.487                            | **65.780**            | **0.344**               | **2.412**               |
>
> For the IU X-ray dataset, our method achieved **the best FID across all dimensions**, demonstrating its superiority. On the PatchGastricADC22 dataset, while our method ranked second, it was only 0.5 points behind the best-performing method (LoRA) Additionally, our method exhibited the **lowest training time and GPU memory usage** on PatchGastricADC22, justifying the slight increase in FID. These results show that our method provides a good trade-off between performance and training efficiency. The new experiments, conducted with a resolution of 384x384, further convey the robustness of our method.

---

### Official Review · Reviewer_ob8y · 2024-11-03

**Soundness:** 2
**Presentation:** 2
**Contribution:** 2
**Rating:** 5
**Confidence:** 5

**Summary:**

The paper introduces SeLoRA (Self-Expanding Low-Rank Adaptation), an extension of the LoRA (Low-Rank Adaptation) technique, designed for the fine-tuning of large diffusion models specifically in medical image synthesis. The core idea is to dynamically expand the rank of low-rank matrices during training, based on a criterion derived from Fisher information. This adaptation is applied selectively across different layers, allowing for a more effective distribution of ranks that aligns with each layer's significance in the model, particularly within the denoising U-Net of the Stable Diffusion framework.

**Strengths:**

1. The paper is well-organized and easy to follow.

2. The idea of adaptive computation rank selection is interesting and highly relevant for using pre-trained models for downstream tasks in a memory efficient fashion

**Weaknesses:**

- While the paper mainly focusses on making a more efficient LoRA design as claimed by authors, there is no Analysis of Training Efficiency in the paper.

- Lack of detailed explanation of the procedure of selecting hyper-parameters needed.

**Questions:**

- How is the proposed method compared to other baselines in terms of max training GPU memory, training speed, and
training time costs? An analysis of these criteria would strengthen the paper.

- The paper mentions thresholds (λ and t) for triggering rank expansion. How sensitive is the method to these hyper-parameters?

- In figure 7, why are the other methods not generating similar xray images? As per my understanding, they should at-least make a similar image like figure 8.

- The proposed method is just compared to methods with rank pruning methods (dylora and adalora), could you mention why there is no comparing with adaptive rank selection papers? as they seem to be a similar approach to your paper.

---

> ### Author Response · Authors · 2024-11-25
> **(1/2) Rebuttal from Authors of Paper10521 to Reviewer ob8y**
>
> We appreciate Reviewer **ob8y**’s valuable and constructive feedback on the analysis of training efficiency and hyper-parameter selection. We have conducted new analyses and additional experiments to address the concerns. Below we address your raised issues point-by-point. Our manuscript is updated accordingly.
>
>
> **Training Efficiency Analysis**
>
> To analyze training efficiency, we conducted experiments for *SeLoRA* and related methods on the IU X-ray and PatchGastricADC22 datasets, recording training time and GPU memory usage every 40 steps. **The results are detailed in Table 8 (training time) and Table 7, Figures 11 and 12 (GPU memory usage) in the updated manuscript**. Key findings are summarized below:
>
> --------------------
> | METHODS   |  GPU Memory Usage (PatchGastricADC22)  | GPU Memory Usage (IU X-Ray) | Training Time (PatchGastricADC22) | Training Time (IU X-Ray) |
> |----------------|-----------------------------------------|----------------------------------|---------------------------------------|-------------------------------|
> | *LoRA*        | 21.591                                  | 20.508                           | 2:40:15                              | 1:44:43                      |
> | *AdaLoRA*     | 20.908                                  | 19.644                           | 2:37:42                              | 1:49:44                      |
> | *DyLoRA*      | 20.921                                  | **18.995**                       | 2:44:01                              | **1:41:30**                  |
> | *SeLoRA*      | **20.751**                              | 19.179                           | **2:36:56**                          | 1:47:01                      |
>
> --------------------
>
> **Key results**:
>
> * *SeLoRA* achieves the best FID for both datasets.
> * *SeLoRA* consistently balances training time, memory usage, and performance.
> * The additional overhead of Fisher Information computation is minimal, making *SeLoRA* efficient even in resource-constrained settings.
>
> Our method achieves the **shortest training time** on the PatchGastricADC22 dataset and ranks third on the IU X-Ray dataset, trailing the fastest method (*DyLoRA*) by only 6 minutes—a negligible difference. When combined with the FID evaluation (Tables 5 and 6), the advantage becomes clear: on the IU X-Ray dataset, our method outperforms *DyLoRA* by 23 FID points. On the PatchGastricADC22 dataset, where our method is the fastest, it incurs only a minor 0.5-point FID disadvantage compared to the best-performing method (*LoRA*). These results highlight a favorable trade-off between training time and performance.
>
> For GPU memory usage (Figures 11 and 12), our method consistently ranks among the most efficient. It achieves the lowest memory usage on the PatchGastricADC22 dataset and ranks second on the IU X-Ray dataset. While slightly behind *DyLoRA* in memory efficiency on the IU X-Ray dataset, the advantage of 23 points in FID justifies this trade-off.
>
> Overall, our method **strikes the best balance** between performance and training efficiency.

---

> ### Author Response · Authors · 2024-11-25
> **(2/2) Rebuttal from Authors of Paper10521 to Reviewer ob8y**
>
> **Hyper-parameter selection**
>
> We appreciate the reviewer’s concerns about hyperparameter selection and provide the following clarifications:
>
> - **$\lambda$ Parameter**: $\lambda$ represents the ratio between the previous and current Fisher Information (FI) scores, typically close to 1. Our analysis shows $\lambda$ = 1.1 is optimal for capturing incremental improvements, as detailed in Section 4.2 (Ablation Study). Even with $\lambda$ = 1.3, performance degradation is minimal (a 9-point FID increase), demonstrating the method's robustness and flexibility.
>
> - **$t$ Parameter**: A larger $t$ ensures reliable rank expansion by evaluating Fisher Information over a meaningful batch size. This prevents unnecessary rank increases and maintains training efficiency.
>
> These clarifications, supported by ablation results, highlight the robustness of our approach to variations in these parameters.
>
>
>
> **Clarification on Noisy Images in Figure 7**
>
> Notably, the _Montgomery County CXR_ dataset has only ~100 training samples. Most *LoRA*-based methods failed due to uniform rank allocation, resulting in noisy outputs. *SeLoRA’s* adaptive rank allocation enabled it to generate meaningful images even in this low-data regime, a critical advantage for medical imaging applications.
>
>
>
> **Comparison to Adaptive Rank Selection Methods**
>
> Adaptive rank selection methods, such as *Generalized LoRA*, rely on computationally expensive evolutionary searches to determine optimal layer-wise rank configurations. Their process involves training ~50 subnets from a supernet, followed by evolutionary search and additional training of selected subnets, significantly increasing training time. As noted in their paper, “training time may increase due to this search process,” and comparisons are often made against hand-tuned hyperparameter configurations for *LoRA*.
>
> In contrast, *SeLoRA* and the methods we benchmarked against (*DyLoRA* and *AdaLoRA*) determine the optimal rank dynamically during a single training run, significantly reducing training time. While adaptive rank selection can yield strong results with extensive computational resources, its design is impractical for resource-constrained environments.
>
> Comparing *SeLoRA* with adaptive rank selection methods would be inherently unfair due to the stark disparity in training time and computational costs. Instead, our work focuses on achieving competitive results within a single training process, aligning with the practical constraints of real-world applications.

---

> > ### Comment · Reviewer_ob8y · 2024-11-25
> > **Response to authors comment on my review.**
> >
> > I thank the authors for addressing my concerns and providing additional experiments. Below, I present my further observations.
> >
> > Regarding efficiency, the comparison of results generally indicates only a minor difference in training time. The authors state that their method ranks third on the IU X-Ray dataset, lagging behind the fastest method (DyLoRA) by just 6 minutes—a difference that can be considered negligible. However, on the GastricADC22 dataset, where the authors claim to be the fastest, their advantage is limited to only 1 minute. Consequently, the results do not strongly support the claimed efficiency of the method as emphasized in the paper.
> >
> > I acknowledge the authors' efforts in addressing other points of feedback and have raised my score accordingly. Nonetheless, I maintain the opinion that the paper falls below the acceptance threshold for ICLR.

---

> > > ### Author Response · Authors · 2024-11-27
> > >
> > > Thank you for raising the score. As discussed during the rebuttal, while *SeLoRA* does not always achieve the highest efficiency, it strikes the best balance between performance and training efficiency. We have carefully refined our arguments in the main text to reflect this. Thank you once again for your constructive feedback.

---

### Author Response · Authors · 2024-11-25
**General Rebuttal**

We thank the reviewers for their thoughtful and constructive feedback. In response, we have thoroughly revised the manuscript and addressed all concerns raised. The main updates include:
* We included detailed metrics on training time, GPU memory usage, and computational costs, demonstrating that *SeLoRA* strikes an excellent balance between efficiency and performance (addressing Reviewer **ob8y**, **7CbA**, **Pa5H**).
* We performed experiments on pathology images using the _PatchGastricADC22_ dataset, showcasing *SeLoRA*'s versatility and competitive performance on a different modality (addressing Reviewer **Ko6A**, **7CbA**, **Pa5H**).
* We conducted additional experiments at 384×384 resolution to demonstrate the robustness of *SeLoRA* to high-resolution scenarios (addressing Reviewer **Ko6A**).
* We adjusted the dataset splits (70%-10%-20%) for training, validation, and testing to ensure robust and fair evaluations (addressing Reviewer **Ko6A**, **7CbA**).
* For larger datasets like MIMIC-CXR, we clarified our choice to focus on smaller datasets, aligning with the challenges of medical image synthesis in resource-constrained settings (addressing Reviewer **Ko6A**, **7CbA**)

These updates, highlighted in blue in the revised manuscript, strengthen our contributions and address all reviewer concerns.

Detailed, point-by-point responses are provided below. We thank the reviewers again for their insights and welcome further discussions.

---

### Meta-Review · Area_Chair_rZoF · 2024-12-19

**Metareview:**

This paper presents a method to use latent diffusion model for medical image synthesis. While the results show some improvement by FID scores, there is a lack of analysis how the synthesis makes an impact for downstream medical image applications. The authors claimed that the proposed method have an excellent trade-off between efficiency and performance. I am not sure how the training efficiency here helps the clinicians. In addition, I am not convinced in some of he claims and results. For example, the authors claimed that “SeLoRA strikes an excellent balance between efficiency and performance”. However, the training efficiency from different algorithms are very close and differs by a few mins with a total training time around 2hours 40 mins or 1 hour 45 mins in two different datasets, as shown in the table from the rebuttal. This is hard for me to accept the claim that the proposed method benefits much on training efficiency. The ablation study shows that the FID scores change abruptly for lamda from 1.0 to 1.1, a detailed analysis shall be required to evaluate the robustness of the results.

**Additional Comments On Reviewer Discussion:**

In the rebuttal, the authors claimed  that “SeLoRA strikes an excellent balance between efficiency and performance”. However, the training efficiency from different algorithms are very close and differs by a few mins with a total training time around 2hours 40 mins or 1 hour 45 mins in two different datasets, as shown in the table from the rebuttal.

---

### Decision · Program_Chairs · 2025-01-22

Reject